# SMS messaging to improve retention and viral suppression in prevention of mother-to-child HIV transmission (PMTCT) programs in Kenya: A 3-arm randomized clinical trial

John Kinuthia[1][☯][‡], Keshet Ronen[2][☯][‡]*, Jennifer A. Unger[2,3], Wenwen Jiang[4], Daniel Matemo[1], Trevor Perrier[5], Lusi Osborn[1], Bhavna H. Chohan[2,6], Alison L. Drake[2], Barbra A. Richardson[2,7], Grace John-Stewart[2,4,8]

1 Department of Research and Programs, Kenyatta National Hospital, Nairobi, Kenya, 2 Department of Global Health, University of Washington, Seattle, Washington, United States of America, 3 Department of Obstetrics and Gynecology, University of Washington, Seattle, Washington, United States of America, 4 Department of Epidemiology, University of Washington, Seattle, Washington, United States of America, 5 Paul G. Allen School of Computer Science and Engineering, University of Washington, Seattle, Washington, United States of America, 6 Kenya Medical Research Institute, Nairobi, Kenya, 7 Department of Biostatistics, University of Washington, Seattle, Washington, United States of America, 8 Department of Pediatrics, University of Washington, Seattle, Washington, United States of America

☯ These authors contributed equally to this work.
‡ These authors share first authorship on this work.
* keshet@uw.edu

**Data Availability Statement:** All data files are available for download from GitHub, at https://github.com/keshetronen/mobile_wach_x_public).

## Abstract

### Background

Pregnant and postpartum women living with HIV (WLWH) need support for HIV and maternal child health (MCH) care, which could be provided using short message service (SMS).

### Methods and findings

We compared 2-way (interactive) and 1-way SMS messaging to no SMS in a 3-arm randomized trial in 6 MCH clinics in Kenya. Messages were developed using the Health Belief Model and Social Cognitive Theory; HIV messages were integrated into an existing MCH SMS platform. Intervention participants received visit reminders and prespecified weekly SMS on antiretroviral therapy (ART) adherence and MCH, tailored to their characteristics and timing. Two-way participants could message nurses as needed. Clinic attendance, viral load (VL), and infant HIV results were abstracted from program records. Primary outcomes were viral nonsuppression (VL ≥1,000 c/ml), on-time clinic attendance, loss to follow-up from clinical care, and infant HIV-free survival. Among 824 pregnant women randomized between November 2015 and May 2017, median age was 27 years, gestational age was 24.3 weeks, and time since initiation of ART was 1.0 year. During follow-up to 2 years postpartum, 9.8% of 3,150 VL assessments and 19.6% of women were ever nonsuppressed, with no significant difference in 1-way versus control (11.2% versus 9.6%, adjusted risk ratio (aRR) 1.02 [95% confidence interval (CI) 0.67 to 1.54], p = 0.94) or 2-way versus control (8.5% versus 9.6%, aRR 0.80 [95% CI 0.52 to 1.23], p = 0.31). Median ART adherence and

**Funding:** This research was funded by the following grants from the National Institutes of Health (https://www.nih.gov): R01HD080460 (to GJS), K24HD054314 (to GJS), P30AI027757 (to GJS), K18MH122978 (to KR), K01AI116298 (to ALD), K12HD001264 (to JAU). The funders had no role in study design, data collection and analysis, decision to publish, or preparation of the manuscript.

**Competing interests:** I have read the journal's policy and the authors of this manuscript have the following competing interests: BR has been on a DSMB and mock FDA advisory panel for Gilead; and GJS reports financial support from NIH. All other authors declare no competing interests.

**Abbreviations:** ANC, antenatal care; aRR, adjusted risk ratio; ART, antiretroviral therapy; CDC, Centers for Disease Control and Prevention; CI, confidence interval; DBS, dried blood spot; DSMB, Data Safety Monitoring Board; EBF, exclusive breastfeeding; EFZ, efavirenz; EMR, electronic medical record; GEE, generalized estimating equations; HR, hazard ratio; IMB, information–motivation–behavior skills; IQR, interquartile range; IVR, interactive voice response; KEMRI, Kenya Medical Research Institute; MCH, maternal child health; Mobile WACh, Mobile Solutions of Women and Children's Health; mHealth, mobile health; NNRTI, non-nucleoside reverse transcriptase inhibitor; NRTI, nucleoside/tide reverse transcriptase inhibitor; ODK, Open Data Kit; OLA, oligonucleotide ligation assay; PCR, polymerase chain reaction; PMTCT, prevention of mother-to-child HIV transmission; py, person-years; RCT, randomized clinical trial; SMS, short message service; TDF, tenofovir; VL, viral load; WLWH, women living with HIV.

incident ART resistance did not significantly differ by arm. Overall, 88.9% (95% CI 76.5 to 95.7) of visits were on time, with no significant differences between arms (88.2% in control versus 88.6% in 1-way and 88.8% in 2-way). Incidence of infant HIV or death was 3.01/100 person-years (py), with no significant difference between arms; risk of infant HIV infection was 0.94%. Time to postpartum contraception was significantly shorter in the 2-way arm than control. Study limitations include limited ability to detect improvement due to high viral suppression and visit attendance and imperfect synchronization of SMS reminders to clinic visits.

## Conclusions

Integrated HIV/MCH messaging did not improve HIV outcomes but was associated with improved initiation of postpartum contraception. In programs where most women are virally suppressed, targeted SMS informed by VL data may improve effectiveness. Rigorous evaluation remains important to optimize mobile health (mHealth) interventions.

## Trial registration

ClinicalTrials.gov number NCT02400671.

## Author summary

### Why was this study done?

- Interactive short message service (SMS) text messaging has the potential to provide remote support and information about HIV care to peripartum women living with HIV (WLWH) in resource-limited settings.

- Prevention of mother-to-child HIV transmission (PMTCT) programs have observed declining retention and treatment adherence over the postpartum period, which results in persistent risk of infant HIV infection.

- Previous research has shown that interactive SMS can improve early retention in perinatal women, but it is unknown whether long-term interactive SMS systems can durably improve retention and viral suppression in PMTCT.

- We hypothesized that a holistic SMS platform that addressed relevant HIV care and maternal child health (MCH) care issues and incorporated comprehensive input from WLWH regarding their preferences would improve HIV care outcomes.

### What did the researchers do and find?

- We integrated messages about HIV treatment, guided by input from WLWH who had previously attended PMTCT programs into an existing interactive SMS platform for MCH.

- We conducted a randomized controlled trial among 824 pregnant WLWH in Kenya, comparing 1-way SMS (in which participants could receive SMS but not respond) and 2-way SMS (in which participants could receive and send SMS to nurses) to no

treatment control. We followed WLWH from pregnancy to 2 years postpartum and evaluated impact of SMS on long-term timely attendance at clinic visits, retention, viral suppression, and infant HIV-free survival.

- We found no significant effect of 1-way or 2-way SMS on HIV viral load (VL) nonsuppression (9.6% in control versus 11.2% in 1-way and 8.5% in 2-way), on-time clinic appointment attendance (88.2% in control versus 88.6% in 1-way and 88.8% in 2-way), and infant HIV or death (2.6/100 person-years (py) in control versus 2.3/100 py in 1-way and 4.1/100 py in 2-way).

## What do these findings mean?

- Integrated SMS messaging on HIV treatment and MCH did not improve HIV outcomes, but the previously reported apparent effect on initiation of postpartum contraception was preserved.

- Study limitations included limited ability to detect outcome improvements due to high treatment success in the control arm and lack of systematic integration of intervention with medical records. In HIV treatment programs that achieve high levels of retention in care and viral suppression, targeted SMS informed by VL data though health record integration may improve effectiveness.

- As mobile health (mHealth) programs continue to gain popularity and increase in scale, rigorous evaluation of clinical effect remains important to optimize interventions.

## Introduction

Mobile health (mHealth) interventions are increasingly deployed to facilitate engagement in care, retention, and adherence to treatment in HIV programs. mHealth systems are attractive as they extend the reach of the clinic, have efficiencies that can support overextended healthcare workers, and can be implemented at relatively low cost. There is promising, but mixed, evidence that short message service (SMS) may improve retention in care and antiretroviral therapy (ART) adherence and viral suppression. A 2020 systematic review identified 27 studies evaluating the effect of SMS, interactive voice response (IVR), and phone calls on HIV treatment in low- and middle-income countries [1]. Of these studies, 41% reported significant positive effect on ART adherence and 21% on retention in care; the authors identified short follow-up time and small sample sizes as limitations. Similarly, a 2019 systematic review and meta-analysis of phone interventions to improve ART adherence identified 9 trials evaluating text message interventions [2]. Meta-analysis identified moderate or nonsignificant effect of text messages on ART adherence, depending on the measure used. Studies with interactive interventions reported stronger effects, as has been reported in other health conditions [3]. mHealth interventions could be especially useful in prevention of mother-to-child HIV transmission (PMTCT) programs [4]. During the peripartum period, women living with HIV (WLWH) need to optimize both maternal child health (MCH) and HIV care. Several studies have found waning ART adherence postpartum and suboptimal viral suppression in cohorts of peripartum women [5–7]. Qualitative studies note that women desire peripartum support and encouragement, both for MCH and for HIV care [8–10].

We previously developed an open-source human–computer hybrid interactive SMS platform (Mobile Solutions of Women and Children's Health (Mobile WACh)) to improve MCH outcomes during pregnancy and early postpartum [11]. Mobile WACh significantly reduced time to uptake of effective contraception (in both a 1-way arm, where participants received SMS but could not respond, and a 2-way arm, where participants could additionally respond and converse with nurses) and prolonged exclusive breastfeeding (EBF) (in the 2-way arm) in previous randomized clinical trials (RCTs) among peripartum women in Kenya [12,13]. We hypothesized that integrating HIV and MCH messages within a single SMS platform would improve both HIV and MCH outcomes among mother–infant pairs in PMTCT programs. Therefore, we developed a new SMS platform (Mobile WACh-X), which added HIV adherence reminder and support messaging to the existing Mobile WACh platform and conducted an RCT to determine the effect of 2-way or 1-way SMS on viral nonsuppression, programmatic retention in PMTCT, ART adherence, and infant HIV-free survival [14].

## Methods

### Study design

This was an individually randomized nonblinded 3-armed trial, conducted at 6 public MCH clinics in Kenya: 2 in peri-urban Nairobi County and 4 in rural Siaya, Kisumu, and Homa Bay Counties. Ethical approval was obtained from the University of Washington and the Kenyatta National Hospital–University of Nairobi. The trial protocol was previously published and registered at ClinicalTrials.gov as NCT02400671 [14].

### Participants

Participants were recruited from antenatal care (ANC) clinics. Eligible participants were pregnant WLWH, ≥14 years old, had daily access to a mobile phone (own or shared) with a Safaricom SIM card, were willing to receive SMS, planned to reside in the area for 2 years postpartum, planned to receive both MCH and HIV care at the facility, and were not enrolled in other studies. During the first 6 months of enrollment, an additional eligibility criterion of ≤36 weeks gestational age was used; this criterion was removed to expedite recruitment. To maximize generalizability, women who were illiterate but had another person to help them read and write SMS were eligible.

Study staff obtained verbal consent to assess eligibility using a tablet-based questionnaire, using Open Data Kit (ODK). Eligible women provided written informed consent to participate. Consent materials were provided in English, Kiswahili, and Dholuo. In accordance with Kenyan regulations, adolescent participants age 14 to 17 years were considered emancipated by pregnancy and provided consent without the need for parental permission. The target sample size for the RCT was 825 participants. The target was reached, but 1 individual was later found to have been enrolled twice. It was determined that this participant was not pregnant at her first enrollment; thus, only data from the second enrollment were retained.

### Randomization and masking

Participants were individually randomized by study nurses 1:1:1 to 1 of 3 arms: control (no SMS), 1-way SMS (participants received SMS but were unable to send messages to the study), and 2-way SMS (participants received SMS and could message the study). Randomization was stratified by site with no more than 399 women from any 1 site. A randomization list was generated by a statistician not involved with the study, using variable block sizes (blocks of 5 different sizes ranging from 3 to 15 each were randomly selected) using Stata 12.1 ralloc. ado

v.3.5.2. Allocation codes were placed in sequentially numbered, sealed, opaque envelopes by site, which were sequentially distributed by study nurses to participants and opened by participants. Study investigators were blinded to block number, size, and sequence. Participants and study investigators were not masked to group assignments.

## Procedures

The study intervention has been previously described [10,14]. Participants randomized to 1-way and 2-way SMS arms received weekly, automated messages from enrollment to 2 years after delivery. Additionally, participants received SMS clinic visit reminders. SMS were sent at the participant's preferred time of day (8 AM, 2 PM, or 8 PM), day of week (Monday to Friday), and language (English, Kiswahili, or Dholuo). Two-way automated messages ended with a question related to the message topic to promote engagement. Participants in the 2-way arm could respond to these or initiate a message to the study nurse at any time. Participant messages were answered by a study nurse within 1 business day.

Formative interviews were conducted with peripartum WLWH, healthcare workers, and partners to inform development of the SMS intervention [8,10,15]. SMS content addressed varied topics, including ART adherence, infant HIV prophylaxis, pregnancy education, birth preparedness, pregnancy and delivery complications, infant health, family planning, and clinic appointment reminders. For every 3 messages related to HIV, 1 message was sent related to non-HIV MCH topics (automated SMS bank available on request) [14]. SMS messages were composed based on the Health Belief Model and Social Cognitive Theory [16,17], designed to provide tailored and actionable education, support, and reminder messages to reinforce health behaviors such as clinic attendance and ART adherence. SMS topics were scheduled according to antenatal/postnatal timing and ART experience, with tailored messaging tracks for adolescents, participants newly initiating ART, and participants who experienced pregnancy loss or infant death [14,18]. Participants who had disclosed their HIV status or had their own phone were given the option to receive SMS containing overt HIV-related language; all other participants received SMS with covert references to HIV [10]. Clinic appointment schedules were abstracted from clinical records and entered into the SMS system by the study team. Visit reminder SMS were sent 3 days before the scheduled appointment date, and congratulatory SMS were sent when visits were attended. SMS were sent 3 and 6 days after scheduled visits were missed. All SMS were delivered free of charge through a reverse-billed short code and managed by study nurses through a custom, semiautomated, open-source web application [11,14]. The system was hosted on a password-protected virtual private server.

Data collection occurred at in-person study visits at enrollment in pregnancy, 6 weeks postpartum, and 6, 12, 18, and 24 months postpartum. At each study visit, questionnaire data were collected using ODK, blood samples were collected and archived, and infant dried blood spots (DBSs) were collected. Study staff abstracted viral load (VL) results, appointments, deliveries, clinic visits, and medication refills from clinical records. All clinical care was provided by clinic staff. Since retention in care was a study outcome, study visit attendance was not optimized by the study team, except at the exit visit. If participants did not attend a study visit between 21 and 25.5 months postpartum, study staff traced the participant by phone and home visit. If participants did not wish to attend their exit visit at the clinic, study staff completed the visit at participants' homes.

At visits where no programmatic HIV VL data were available, the study performed VL assays on archived plasma samples. These VL assays were performed at the Kenya Medical Research Institute (KEMRI)/Centers for Disease Control and Prevention (CDC) in Kisumu or Nairobi, Kenya, using the Roche COBAS TaqMan Analyzer or COBAS TaqMan Version 2.0

(CAP/CTMv2.0) platform. Infant HIV DNA polymerase chain reaction (PCR) testing were abstracted from routine PMTCT program at 6 weeks, 6 months, and 12 months. The study performed an additional infant test at study exit using fourth-generation rapid or fourth-generation ELISA tests using DBS. Maternal plasma ART resistance was assessed using an oligonucleotide ligation assay (OLA) on archived enrollment plasma samples and subsequent samples where VL was nonsuppressed. [19] OLA is designed to detect mutations at HIV-1 pol reverse transcriptase codons 103, 181, 184, and 190 that can confer resistance to non-nucleoside reverse transcriptase inhibitors (NNRTIs) and nucleoside/tide reverse transcriptase inhibitors (NRTIs). If samples yielded indeterminate results by OLA, the amplicon from RNA also underwent consensus sequencing.

## Outcomes

Primary trial outcomes were maternal virologic nonsuppression, on-time visit attendance, loss to follow-up, and infant HIV infection or death. Secondary outcomes were maternal ART adherence and ART resistance. MCH outcomes previously noted to be improved by the Mobile WACh platform (timing of postpartum contraception and EBF duration) were also assessed. Qualitative evaluation of participant experiences will be reported elsewhere. Virologic nonsuppression was defined as plasma VL $\geq$1,000 copies/mL. A secondary analysis evaluated a cutoff of the assay limit of detection (20 copies/mL for plasma, 839 copies/mL, or 550 copies/mL for DBS depending on assay type). In order to probe potential differences in effect in particular groups of women, subgroup analyses for nonsuppression based on participant age, phone sharing, pregnancy history, employment, HIV status disclosure, education, VL, and ART resistance at baseline were conducted. On-time attendance was defined as attending clinic within 2 weeks of (before or after) a scheduled appointment. Loss to follow-up was defined as not attending any clinic visits for at least 6 months. A combined outcome of infant HIV infection or death was based on clinic records, HIV testing, and participant reports of death. ART adherence was defined as the proportion of days "covered" by ART between pharmacy refills. ART resistance was defined as detection of any resistance mutations by OLA at an abundance of 10% or more.

Adverse events, including pregnancy loss, hospitalization, death, and social harms such as HIV status disclosure, intimate partner violence, or loss of housing, were monitored. Serious adverse events (maternal or infant hospitalization or death) were reported to the Kenyatta National Hospital/University of Nairobi ethics review committee. All events were reported to the University of Washington institutional review board annually and to the trial Data Safety Monitoring Board (DSMB) at annual meetings.

## Statistical analysis

Our analysis compared outcomes between each intervention arm and the control arm and between the combination of either intervention arm and control. Trial sample size was calculated for 80% power to detect a hazard ratio (HR) of <0.65 for either viral nonsuppression or loss to follow-up, assuming a 25/100 py incidence of each outcome during 2-year follow-up [14]. This sample size was estimated to have power to detect an HR of <0.55 for drug resistance (assuming incidence of 15/100 py) and HR >2.0 for HIV-free survival (assuming incidence of 10/100 py). The initial analysis approach (time to event) only utilized data until first viral nonsuppression and required initial viral suppression at baseline. Because WLWH may have episodic nonadherence and viral nonsuppression and because we had serial VL data, generalized estimating equations (GEE) that incorporate all VL data increased study power to detect effect of the intervention. We adapted our prespecified analysis approach

(NCT02400671) accordingly. Using repeated measures approaches, the study would have power to detect a change from baseline of 70% to 85% prevalence of viral suppression to 95% suppression in intervention arms with <168 women per arm, with fewer women per arm depending on the number of repeated measures per woman. All analyses were intent to treat, adjusted for baseline differences between arms in participant characteristics (primigravida and employment status).

**Viral nonsuppression.** Prevalence of viral nonsuppression during follow-up was defined as the proportion of VL measurements that were nonsuppressed at any time in follow-up and compared using GEE clustered by participant, with log-binomial link, exchangeable correlation structure, and robust standard errors. Data were used after enrollment and ≥4 months since ART initiation, to include only data after VL could have been suppressed by ART. Visits with nonsuppressed VL within 30 days of a previous nonsuppressed visit were excluded. Secondary analyses were conducted comparing the incidence of first virologic nonsuppression by Cox regression, comparing the incidence of virologic suppression as a recurrent event by Andersen–Gill regression with robust error estimation clustered by participant, and comparing the cumulative incidence of nonsuppression among women with any post-enrollment VL data by delivery, 6, 12, and 24 months postpartum by log-binomial regression. Time of event was calculated as the midpoint between the last suppressed and the first unsuppressed VL. The prespecified primary analysis was modified between publication of the trial protocol [14] and conducting the analysis: GEE was used as the primary analysis in place of Cox regression, as explained above. Additional exploratory analyses included comparisons of the 2-way versus 1-way arm and analyses using different VL thresholds (undetectable versus detectable using limit of detection).

**Retention and engagement in care.** On-time clinic visit attendance during follow-up to 12 and 24 months postpartum was defined as the proportion of scheduled clinic visits attended on time and compared using GEE clustered by participant, with Poisson link, robust standard errors, and exchangeable correlation structure. The proportion of women lost to follow-up at 12 and 24 months postpartum were compared between arms using log-binomial regression. Time to loss to follow-up was compared between arms using Cox regression for the 0 to 365 day and 0 to 730 day intervals. An exploratory analysis was conducted among intervention participants, comparing on-time attendance of visits that successfully received an SMS visit reminder within the preceding 2 weeks versus visits that did not due to system errors.

**Infant HIV-free survival.** Time to infant HIV infection or death after live birth in the first 850 days postpartum was compared using Cox regression.

**ART adherence.** We classified ART adherence as high (≥95%) versus low (<95%) based on 2 approaches: pharmacy refills and self-report. Pharmacy refill data were abstracted from records of ART doses administered during HIV care visits. Longitudinal ART adherence was calculated as the proportion of days between visits that were covered by doses dispensed. Self-reported data were based on the number of doses missed in the last 30 days at study visits. For each method of ascertainment, GEE log-binomial regression with exchangeable correlation structure and robust standard error was used to compare the prevalence of high (≥95%) versus low (<95%) adherence between randomization arms.

**ART resistance.** Time to development of ART resistance was compared between randomization arms using Cox regression on data from enrollment until 850 days postpartum. Plasma samples with VL <1,000 copies/mL were not tested for resistance and were assumed to not have resistance mutations.

**MCH outcomes.** Since Mobile WACh-X messaging was derived from Mobile WACh messaging, which focused on MCH topics and was shown to prolong EBF and increase contraception uptake in HIV–negative women [12,13], we explored whether efficacy of MCH

messaging persisted with addition of HIV messages. Time to contraception initiation by 6, 12, and 24 months postpartum and time to introduction of complementary foods by 6 months postpartum were compared between arms by Cox regression.

## Results

Between November 22, 2015 and May 4, 2017, 9,697 women attended ANC at study sites, of whom 1,280 (13.4%) were living with HIV and were referred for eligibility screening (Fig 1). Of 900 eligible women, 824 (91.6%) were enrolled and randomized, 271 to 1-way SMS, 276 to 2-way SMS, and 277 to the control arm. All participants received their allocated intervention. Twenty-six participants discontinued the SMS intervention (12 in 1-way and 14 in 2-way); 4 participants withdrew (2 in 1-way and 2 in 2-way); and 71 (8.6%) participants did not attend their exit visit (Fig 1). The trial was carried out to completion.

### Baseline participant characteristics

Enrollment characteristics are summarized in Table 1. Each site contributed 7.6% to 27.7% of the study population. Median age was 27 years (interquartile range (IQR) 23 to 31), and median gestational age was 24.3 weeks (IQR 18.3 to 29.6). Median monthly household income was 8,000 KES or 80 USD (IQR 40 to 150), most women (84.3%) were married or cohabiting, 19.7% had <8 years of education, 50.9% were employed, and 14.0% were primigravida. Phone sharing was reported by 29.7% of women, and 94.1% reported they could read and write SMS

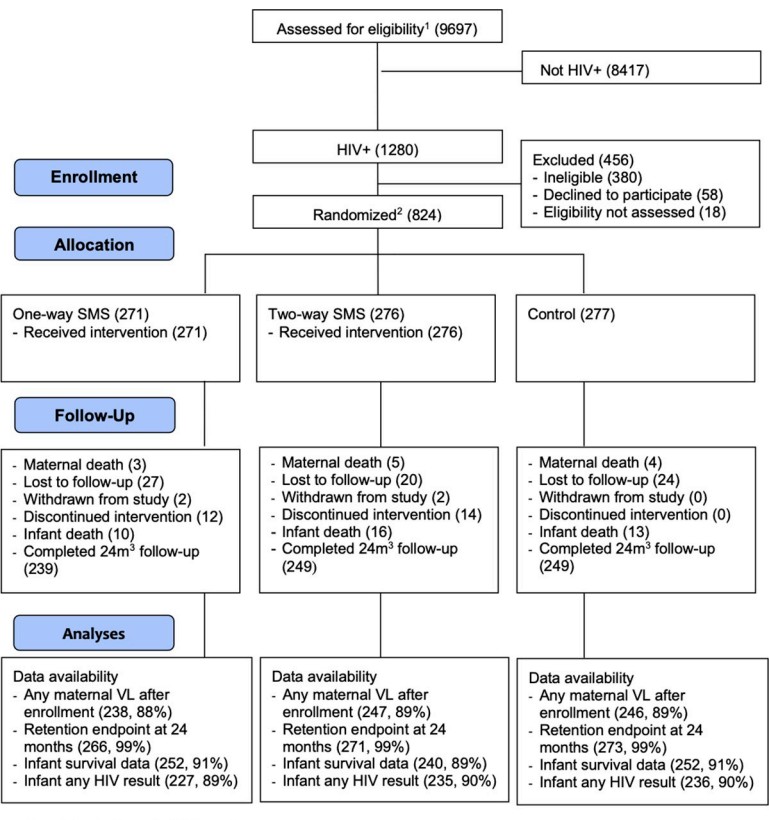

**Fig 1. Participant flowchart. MCH, maternal child health; SMS, short message service; VL, viral load.**

**Table 1. Participant baseline characteristics.**

| | | Overall | | Control | | One-way | | Two-way |
|---|---|---|---|---|---|---|---|---|
| | | | | **n (%) or median (IQR)** | | | | |
| *Sociodemographic* | | | | | | | | |
| Age (years) | 824 | 27 (23–31) | 277 | 27 (23–31) | 271 | 28 (23–30) | 276 | 27 (23–31) |
| <8 years of education | 822 | 162 (19.7%) | 277 | 54 (19.5%) | 269 | 54 (20.1%) | 276 | 54 (19.6%) |
| Monthly household income (KES) | 527 | 8,000 (4,000–15,000) | 177 | 7,000 (4,000–15,000) | 171 | 7,000 (3,200–15,000) | 179 | 9,000 (3,000–15,000) |
| Married/cohabiting | 823 | 694 (84.2%) | 276 | 233 (84.4%) | 271 | 230 (84.9%) | 276 | 231 (83.7%) |
| Employed | 822 | 418 (50.7%) | 276 | 159 (57.4%) | 271 | 126 (46.5%) | 275 | 133 (48.2%) |
| Shares phone | 824 | 245 (29.7%) | 277 | 87 (31.4%) | 271 | 77 (28.4%) | 276 | 81 (29.3%) |
| Can read SMS unassisted | 824 | 808 (98.1%) | 277 | 273 (98.6%) | 271 | 264 (97.8%) | 276 | 271 (97.8%) |
| Can write SMS unassisted | 824 | 775 (94.1%) | 277 | 260 (93.9%) | 271 | 261 (96.3%) | 276 | 254 (92.0%) |
| *Obstetric* | | | | | | | | |
| Primigravida | 824 | 115 (14.0%) | 277 | 26 (9.4%) | 271 | 40 (14.8%) | 276 | 49 (17.8%) |
| Pregnancy intended | 821 | 452 (55.1%) | 275 | 149 (54.2%) | 271 | 145 (53.5%) | 275 | 158 (57.5%) |
| Gestational age (weeks) | 817 | 24.3 (18.3–29.6) | 274 | 24.7 (19.8–30.3) | 271 | 24.6 (18.1–28.9) | 272 | 23.3 (17.9–29.6) |
| *HIV/ART* | | | | | | | | |
| Time since HIV diagnosis (years) | 820 | 2.00 (0.08–5.00) | 242 | 2.00 (0.08–5.00) | 271 | 2.00 (0.08–5.00) | 275 | 2.00 (0.08–5.00) |
| On ART | 824 | 764 (92.7%) | 277 | 258 (93.1%) | 271 | 249 (91.9%) | 276 | 257 (93.1%) |
| Time since ART start (years) | 819 | 1.00 (0.02–3.21) | 274 | 0.62 (0.01–2.64) | 270 | 1.24 (0.04–3.36) | 275 | 1.10 (0.01–3.42) |
| On ART ≥4 months | 824 | 470 (57.0%) | 277 | 149 (53.8%) | 271 | 161 (59.4%) | 276 | 160 (58.0%) |
| HIV status disclosed to partner | 806 | 662 (81.1%) | 271 | 218 (80.4%) | 264 | 224 (84.8%) | 271 | 220 (81.2%) |
| ART regimen | 771 | | 251 | | 259 | | 261 | |
| AZT + 3TC + NVP | | 45 (5.8%) | | 12 (4.8%) | | 15 (5.8%) | | 18 (6.9%) |
| AZT + 3TC + EFV | | 4 (0.5%) | | 1 (0.4%) | | 3 (1.2%) | | 0 (0.0%) |
| TDF + 3TC + LPV/r | | 12 (1.6%) | | 4 (1.6%) | | 1 (0.4%) | | 7 (2.7%) |
| TDF + 3TC + NVP | | 107 (13.9%) | | 35 (13.9%) | | 35 (13.5%) | | 37 (14.2%) |
| TDF + 3TC + EFV | | 566 (73.4%) | | 190 (75.7%) | | 190 (73.4%) | | 186 (71.3%) |
| TDF + FTC + EFV | | 3 (0.4%) | | 1 (0.4%) | | 2 (0.8%) | | 0 (0.0%) |
| Other | | 34 (4.4%) | | 8 (3.2%) | | 13 (5.0%) | | 13 (5.0%) |
| VL ≥1,000 at enrollment | | | | | | | | |
| Total | 824 | 244 (29.6%) | 277 | 81 (29.2%) | 271 | 75 (27.7%) | 276 | 88 (31.9%) |
| Established ART (≥4 months) | 470 | 61 (13.0%) | 149 | 15 (10.1%) | 161 | 21 (13.0%) | 160 | 25 (15.6%) |
| New ART (<4 months) | 349 | 180 (51.6%) | 125 | 64 (51.2%) | 109 | 53 (48.6%) | 115 | 63 (54.8%) |
| CD4 at enrollment | 697 | 473 (333–632) | 235 | 454 (334–601) | 229 | 495 (350–662) | 233 | 456 (306–612) |
| Adherence characteristics | | | | | | | | |
| IMB score | 551 | 77.3 (73.3–82.7) | 188 | 77.3 (73.3–82.7) | 180 | 77.3 (72.0–82.7) | 183 | 77.3 (73.3–82.7) |
| Information (ART literacy) | 695 | 85.0 (75.0–95.0) | 234 | 85.0 (75.0–95.0) | 226 | 80.0 (75.0–95.0) | 235 | 85.0 (80.0–95.0) |
| Motivation (to take ART) | 695 | 60.0 (40.0–80.0) | 232 | 60.0 (45.0–80.0) | 227 | 60.0 (40.0–80.0) | 236 | 60.0 (45.0–80.0) |
| Behavior skills (for adherence) | 584 | 80.0 (77.1–91.4) | 202 | 80.0 (77.1–90.7) | 191 | 82.9 (77.1–94.3) | 191 | 80.0 (75.7–88.6) |

ART, antiretroviral therapy; AZT, Zidovudine; EFV, Efavirenz, IMB, information–motivation–behavior skills; IQR, interquartile range; LPV/r, Lopinavir/Ritonavir; NVP, Nevirapine; SMS, short message service; TDF, tenofovir; VL, viral load.

unassisted. Median time since HIV diagnosis was 2 years (IQR 0.08 to 5.00), and 92.7% of women were on ART at enrollment with a median time since ART initiation of 1.00 year (IQR 0.02 to 3.21). Almost all women were on NNRTI regimens, the majority of which were efavirenz (EFZ) based (73.4%), and most (89.2%) received tenofovir (TDF)-containing regimens. At enrollment, 29.6% of women had VLs ≥1,000 copies/mL: 51.6% of women who initiated ART within the past 4 months and 13.0% of women who had been on ART for at least 4 months.

The median CD4 count was 473 cells/mL (IQR 333 to 632). Characteristics were balanced by randomization arm, with the exception of employment (higher in control than intervention arms) and primigravida (lower in control than intervention arms).

## Outcomes of RCT

Overall, 727 participants had ≥1 VL available at ≥4 months after ART initiation, for a total of 3,150 VL measurements included in analyses. Number of VL measurements per participant did not differ significantly by arm (median 6 [IQR 4 to 7] in all 3 arms). The overall proportion of VL measurements that were nonsuppressed at any time during follow-up was 9.78%: 9.62% in the control arm, 11.20% in the 1-way arm, and 8.53% in the 2-way arm (Table 2). There were no significant differences in the frequency of nonsuppression over all follow-up time in the 1-way versus control arm (adjusted risk ratio (aRR): 1.02 [0.67 to 1.54], $p = 0.94$) or 2-way versus control arm (aRR: 0.80 [0.52 to 1.23], $p = 0.31$). By delivery, 6 months, 12 months, and 2 years postpartum, 11.1%, 12.1%, 16.0%, and 19.6% of women had had virologic nonsuppression, respectively. Cumulative incidence of virologic nonsuppression by 24 months postpartum did not differ between intervention and control: The control arm 19.9% of control, 23.3% of 1-way, and 15.6% of 2-way participants experienced nonsuppression (1-way adjusted Hazard Ratio (aHR): 1.17 [0.83 to 1.65], $p = 0.38$; 2-way aRR: 0.78 [0.53 to 1.15], $p = 0.21$) (Table 2). Incidence of first or recurrent nonsuppression did not differ between intervention and control (1-way aHR for initial nonsuppression: 1.17 (0.82 to 1.68), $p = 0.39$; 0.85 (0.58 to 1.24), $p = 0.39$) (Table 2, Fig 2A, and S1 Table). In exploratory analysis, there was a lower cumulative incidence of viral nonsuppression at 6 and 24 months postpartum in 2-way versus 1-way (24 month aRR: 0.67 [0.45 to 0.99], $p = 0.04$) (Table 2). Secondary analyses using different VL cutoffs yielded similar findings.

A total of 794 participants had ≥1 clinic appointment scheduled during study follow-up, with 12,437 scheduled visits included in analysis. Number of scheduled visits per participant did not differ significantly by arm (median 17 [IQR 11 to 23], 18 [IQR 11 to 23], and 18 [IQR 10 to 23] in control, 1-way, and 2-way arms, respectively). During follow-up to 12 months postpartum, 88.5% of visits were attended within 2 weeks of scheduled dates, and the same proportion up to 24 months postpartum. There was no significant difference between arms in on-time attendance up to 24 months postpartum (1-way aRR 1.00 [0.98 to 1.03], $p = 0.81$; 2-way aRR 1.01 [0.99 to 1.04], $p = 0.36$). In explanatory analysis to determine whether lack of effect was due to SMS system failure, among 3,232 scheduled visits in the intervention arms which had a system-sent reminder SMS within 14 days of visit, 2,914 (90.2%) were attended on time, which was significantly higher than among scheduled visits that did not have an SMS sent within 14 days of visit (4,732/5,458, 86.7%, $p < 0.001$). Incidence of loss to follow-up was 13.6/100 py, with no significant difference between control (12.9/100 py), 1-way (13.8/100 py, aHR: 0.99 [0.69 to 1.42], $p = 0.97$), and 2-way (14.0/100 py, aHR: 1.02 [0.71 to 1.47], $p = 0.90$) (Table 2 and Fig 2B).

A total of 744 participants had live births with known infant survival status at exit, and 705 had at least 1 infant HIV result. Of 744 live-born infants, 7 infants (0.94%) acquired HIV, and 38 infants (5.11%) died by 24 months postpartum. Incidence of infant HIV or death was 3.0/100 py: 2.6/100 py in the control arm, 2.3/100 py in the 1-way arm, and 4.1/100 py in the 2-way arm. There were no significant differences in infant HIV acquisition or death between intervention versus control (1-way aHR: 0.83 [0.37 to 1.87], $p = 0.66$; 2-way aHR: 1.44 [0.71 to 2.92], $p = 0.31$) (Table 2 and Fig 2C).

A total of 759 participants had ≥1 ART pharmacy refill visit recorded, for a total of 12,935 visits (Table 3). The median overall estimated adherence by pharmacy refill was 100% (IQR

**Table 2. Effect of Mobile WACh-X on primary outcomes.**

| | Overall (N = 824) | Control (N = 277) | One-way (N = 271) | Two-way (N = 276) | One-way vs. control | Two-way vs. control | Either arm vs. control | Two-way vs. one-way[a] |
|---|---|---|---|---|---|---|---|---|
| **Virologic nonsuppression[1]** | | | | | | | | |
| GEE model | Frequency of post-enrollment VL NS n unsuppressed/n VL measurements (%) | | | | RR (95% CI), p-value | | | |
| N participants | 727 | 243 | 235 | 249 | | | | |
| VL NS at any time | 308/3,150 (9.78) | 99/1,029 (9.62) | 119/1,066 (11.20) | 90/1,055 (8.53) | cRR: 1.09 (0.73–1.65), p = 0.67 aRR: 1.02 (0.67–1.54), p = 0.94 | cRR: 0.87 (0.56–1.34), p = 0.53 aRR: 0.80 (0.52–1.23), p = 0.31 | cRR: 0.98 (0.68–1.41), p = 0.91 aRR: 0.91 (0.63–1.31), p = 0.60 | cRR: 0.79 (0.52–1.22), p = 0.29 aRR: 0.79 (0.52–1.21), p = 0.28 |
| Log-binomial model | Cumulative incidence of VL NS n ever unsuppressed/n women[b] (%) | | | | | | | |
| Participants with VL NS by 6 months postpartum | 62/495 (12.5) | 20/159 (12.6) | 29/170 (17.1) | 13/166 (7.8) | cRR: 1.36 (0.81–2.26); p = 0.24 aRR: 1.36 (0.82–2.26); p = 0.23 | cRR: 0.62 (0.32–1.20); p = 0.16 aRR: 0.63 (0.33–1.20); p = 0.16 | cRR: 0.99 (0.62–1.59); p = 0.98 aRR: 1.00 (0.63–1.58); p = 1.00 | cRR: 0.46 (0.25–0.86); **p = 0.01** aRR: 0.46 (0.25–0.86); **p = 0.01** |
| Participants with VL NS by 12 months postpartum | 101/633 (16.0) | 32/208 (15.4) | 42/216 (19.4) | 27/209 (12.9) | cRR: 1.26 (0.83–1.93); p = 0.28 aRR: 1.26 (0.83–1.92); p = 0.28 | cRR: 0.84 (0.52–1.35); p = 0.47 aRR: 0.84 (0.52–1.35); p = 0.47 | cRR: 1.06 (0.72–1.55); p = 0.78 aRR: 1.05 (0.72–1.55); p = 0.79 | cRR: 0.66 (0.43–1.04); p = 0.07 aRR: 0.67 (0.43–1.04); p = 0.07 |
| Participants with VL NS by 24 months postpartum | 134/683 (19.6) | 46/231 (19.9) | 53/227 (23.3) | 35/225 (15.6) | cRR: 1.17 (0.83–1.65); p = 0.36 aRR: 1.17 (0.83–1.65); p = 0.38 | cRR: 0.78 (0.53–1.15); p = 0.21 aRR: 0.78 (0.53–1.15); p = 0.21 | cRR: 0.98 (0.72–1.33); p = 0.89 aRR: 0.97 (0.72–1.33); p = 0.87 | cRR: 0.67 (0.45–0.98); **p = 0.04** aRR: 0.67 (0.45–0.99); **p = 0.04** |
| Cox model | | | | | HR (95% CI), p-value | | | |
| n events | 171 | 57 | 63 | 51 | cHR: 1.21 (0.84–1.73); p = 0.31 aHR: 1.17 (0.82–1.68); p = 0.39 | cHR: 0.86 (0.59–1.26); p = 0.45 aHR: 0.85 (0.58–1.24); p = 0.39 | cHR: 1.02 (0.75–1.41); p = 0.88 aHR: 1.00 (0.73–1.38); p = 1.00 | cHR: 0.72 (0.49–1.04); p = 0.08 aHR: 0.72 (0.50–1.05); p = 0.09 |
| n py | 1,596.2 | 547.0 | 490.7 | 558.4 | | | | |
| IR/100 py | 10.7 | 10.4 | 12.8 | 9.1 | | | | |
| **Timely attendance[2]** | | | | | | | | |
| | Proportion of scheduled PMTCT program appointments attended within 2 weeks n attended/n scheduled (%) | | | | RR (95% CI), p-value | | | |
| N participants | 794 | 268 | 264 | 262 | | | | |
| Attendance by 12 months postpartum | 6,885/7,776 (88.5) | 2,306/2,614 (88.2) | 2,297/2,592 (88.6) | 2,282/2,570 (88.8) | cRR: 1.00 (0.98–1.03); p = 0.85 aRR: 1.01 (0.98–1.04); p = 0.52 | cRR: 1.00 (0.98–1.03); p = 0.84 aRR: 1.01 (0.98–1.04); p = 0.55 | cRR: 1.00 (0.98–1.03); p = 0.82 aRR: 1.01 (0.99–1.03); p = 0.47 | cRR: 1.00 (0.97–1.03); p = 0.98 aRR: 1.00 (0.97–1.03); p = 0.99 |
| Attendance by 24 months postpartum | 11,006/12,437 (88.5) | 3,709/4,199 (88.3) | 3,633/4,124 (88.1) | 3,664/4,114 (89.1) | cRR: 1.00 (0.97–1.02); p = 0.78 aRR: 1.00 (0.98–1.03); p = 0.81 | cRR: 1.00 (0.98–1.03); p = 0.70 aRR: 1.01 (0.99–1.04); p = 0.36 | cRR: 1.00 (0.98–1.02); p = 0.95 aRR: 1.01 (0.99–1.03); p = 0.50 | cRR: 1.01 (0.98–1.03); p = 0.50 aRR: 1.01 (0.98–1.03); p = 0.49 |
| **Loss to follow-up[3]** | | | | | | | | |
| | | | | | HR (95% CI), p-value | | | |
| N participants | 813 | 276 | 270 | 267 | | | | |
| n events | 183 | 58 | 62 | 63 | cHR: 1.08 (0.75–1.54); p = 0.68 aHR: 0.99 (0.69–1.42); p = 0.97 | cHR: 1.11 (0.77–1.58); p = 0.58 aHR: 1.02 (0.71–1.47); p = 0.90 | cHR: 1.09 (0.80–1.49); p = 0.58 aHR: 1.01 (0.74–1.38); p = 0.96 | cHR: 1.03 (0.72–1.46); p = 0.89 aHR: 1.03 (0.73–1.47); p = 0.86 |
| n py | 1,349.0 | 450.7 | 447.8 | 450.5 | | | | |
| IR/100 py | 13.6 | 12.9 | 13.8 | 14.0 | | | | |
| **Infant HIV acquisition or death[4]** | | | | | | | | |
| | | | | | HR (95% CI), p-value | | | |
| N with live births and outcome data | 744 | 252 | 240 | 252 | cHR: 0.89 (0.40–1.98); p = 0.77 aHR: 0.83 (0.37–1.87); p = 0.66 | cHR: 1.57 (0.78–3.15); p = 0.20 aHR: 1.44 (0.71–2.92); p = 0.31 | cHR: 1.23 (0.65–2.36); p = 0.53 aHR: 1.14 (0.60–2.20); p = 0.69 | cHR: 1.76 (0.84–3.68); p = 0.13 aHR: 1.73 (0.83–3.62); p = 0.15 |
| n events | 44 | 13 | 11 | 20 | | | | |
| n py | 1,460.7 | 501.8 | 475.8 | 483.0 | | | | |
| IR/100 py | 3.0 | 2.6 | 2.3 | 4.1 | | | | |

(Continued)

**Table 2.** (Continued)

| | Overall (N = 824) | Control (N = 277) | One-way (N = 271) | Two-way (N = 276) | One-way vs. control | Two-way vs. control | Either arm vs. control | Two-way vs. one-way[a] |
|---|---|---|---|---|---|---|---|---|
| ***Virologic nonsuppression*[1]** | | | | | | | | |
| *n* HIV+ | 7 | 1 | 1 | 5 | | | | |
| *n* py | 1,013.6 | 349.4 | 326.4 | 337.9 | | | | |
| IR/100 py | 0.7 | 0.3 | 0.3 | 1.5 | | | | |
| *n* deaths | 38 | 12 | 10 | 16 | | | | |
| *n* py | 1,492.2 | 509.3 | 486.4 | 496.6 | | | | |
| IR/100 py | 2.6 | 2.4 | 2.1 | 3.2 | | | | |

[1]Frequency of VL NS was compared by GEE log-binomial regression. VL NS detected multiple times within 30 days is only counted once.

[2]Proportion of on-time attendance was compared by GEE Poisson regression with exchangeable correlation structure. Only data abstracted from paper records were used, since the intervention would not be expected to affect on-time attendance in a system where the visit schedule was not accessed in real time by study nurses. Scheduled appointments upto 351 or 716 days postpartum were included in the 1-year and 2-year analyses, respectively.

[3]Incidence of LTFU compared by Cox proportional hazards regression. Both paper and EMRs data were included.

[4]Events until 850 days postpartum were included. Analyzed using Cox proportional hazards regression. Infection time = midpoint between date of testing and last negative or delivery (if no previous negative). Death time = exact death date. Event time = earlier of infection time and death time.

[a]Exploratory analysis.

[b]Denominator is cumulative number of women with any post-enrollment VL data within the specified time period.

aHR, hazard ratio adjusted for primigravida and employment; aRR, risk ratio adjusted for primigravida and employment; cHR, crude hazard ratio; CI, confidence interval; cRR, crude relative risk; EMR, electronic medical record; GEE, generalized estimating equations; HR, hazard ratio; LTFU, lost to follow-up; PMTCT, prevention of mother-to-child HIV transmission; py, person-years; RR, relative risk; VL, viral load; VL NS, virologic nonsuppression; IR, incidence rate.

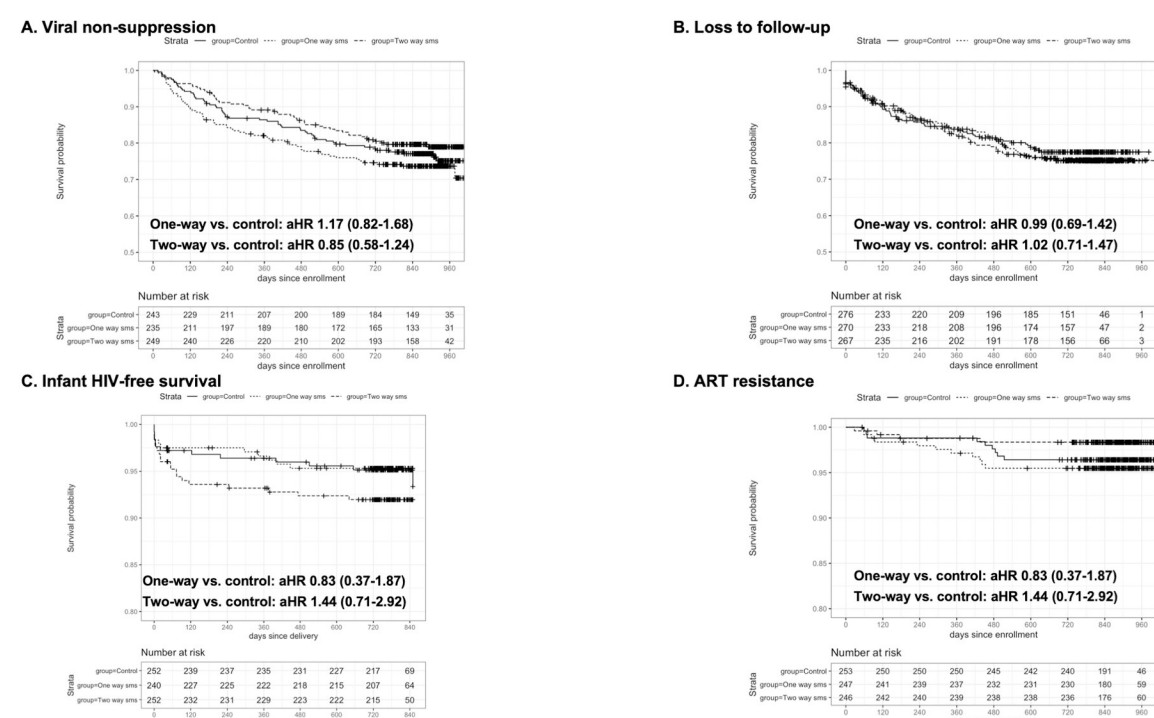

**Fig 2. Effect of Mobile WACh-X on viral nonsuppression, loss to follow-up, infant HIV-free survival, and ART resistance.** aHR, adjusted hazard ratio; ART, antiretroviral therapy; SMS, short message service.

**Table 3. Effect of Mobile WACh-X on secondary outcomes.**

| | Overall (N = 824) | Control (N = 277) | One-way (N = 271) | Two-way (N = 276) | One-way vs. control | Two-way vs. control | Either arm vs. control | Two-way vs. one-way[a] |
|---|---|---|---|---|---|---|---|---|
| | n (%) | | | | RR (95% CI), p-value | | | |
| *Maternal ART adherence: pharmacy refills* | | | | | | | | |
| | Proportion of refill visits with ≥95% adherence n adherent/n visits (%) | | | | | | | |
| N participants | 759 | 256 | 250 | 252 | | | | |
| Median adherence | 100 (97.2–105) | 100 (98.2–107) | 100 (98.4–105) | 100 (96.8–104) | | | | |
| Visits with ≥95% adherence | 8,762/12,395 (70.7) | 2,956/4,191 (70.5) | 2,881/4,096 (70.3) | 2,925/4,108 (71.2) | cRR: 0.99 (0.95–1.04); p = 0.85 aRR: 1.00 (0.96–1.04); p = 0.98 | cRR: 1.00 (0.96–1.05); p = 0.91 aRR: 1.01 (0.97–1.05); p = 0.69 | cRR: 1.00 (0.96–1.04); p = 0.96 aRR: 1.00 (0.97–1.04); p = 0.82 | cRR: 1.01 (0.97–1.05); p = 0.76 aRR: 1.01 (0.97–1.05); p = 0.67 |
| *Maternal ART adherence: self-reported doses missed in the last 30 days* | | | | | | | | |
| | Proportion of visits with ≥95% self-reported adherence n adherent/n visits (%) | | | | | | | |
| N participants | 737 | 248 | 244 | 245 | | | | |
| Mean adherence | 100 (100–100) | 100 (100–100) | 100 (100–100) | 100 (100–100) | | | | |
| Visits with ≥95% adherence | 2,240/2,335 (95.9) | 727/758 (95.9) | 755/785 (96.2) | 758/792 (95.7) | cRR: 1.00 (0.98–1.03); p = 0.83 aRR: 1.00 (0.98–1.03); p = 0.84 | cRR: 1.00 (0.97–1.02); p = 0.79 aRR: 1.00 (0.97–1.02); p = 0.74 | cRR: 1.00 (0.98–1.02); p = 0.97 aRR: 1.00 (0.98–1.02); p = 0.93 | cRR: 0.99 (0.97–1.02); p = 0.64 aRR: 0.99 (0.97–1.02); p = 0.60 |
| *Maternal ART resistance* | | | | | | | | |
| | | | | | HR (95% CI), p-value | | | |
| N participants | 748* | 254 | 248 | 246 | | | | |
| Any mutation post-enrollment | 24 | 9 | 11 | 4 | cHR: 1.28 (0.53–3.08); p = 0.59 aHR: 1.18 (0.49–2.87); p = 0.71 | cHR: 0.46 (0.14–1.49); p = 0.20 aHR: 0.42 (0.13–1.37); p = 0.15 | cHR: 0.87 (0.38–1.98); p = 0.73 aHR: 0.80 (0.35–1.83); p = 0.59 | cHR: 0.36 (0.11–1.13); p = 0.08 aHR: 0.35 (0.11–1.11); p = 0.08 |
| Py | 1,788.42 | 608.29 | 585.33 | 594.80 | | | | |
| Incidence rate/100 py | 1.34 | 1.48 | 1.88 | 0.67 | | | | |

Adjusted models include primigravida and employment at baseline.

*Resistance was assessed post-enrollment, and women with baseline resistance were excluded. Women with suppressed viral load were assumed to have no resistance.

aHR, adjusted hazard ratio; aRR, adjusted relative risk; ART, antiretroviral therapy; CI, confidence interval; cHR, crude hazard ratio; CI, confidence interval; cRR, crude relative risk; HR, hazard ratio; py, person-years.

97.2 to 105.0). Among all clinic visits, 8,762 visits (70.7%) were attended soon enough after the preceding visit that ≥95% of days in the interval were covered by ART doses dispensed at the preceding visit. There were no significant differences between intervention and control (1-way aRR of high adherence: 1.00 [0.96 to 1.04], p = 0.98; 2-way aRR: 1.01 [0.97 to 1.05], p = 0.69). A total of 737 participants had self-reported ART adherence data from 2,335 study visits. Median ART self-reported adherence was 100% (IQR 100 to 100), and self-reported ART adherence ≥95% was reported at a total of 2,240 visits (95.9%), with no significant differences by arm (Table 3).

Of 748 participants without ART resistance at enrollment, 282 (37.7%) qualified for resistance testing. Among these participants, 24 developed an ART resistance mutation, for an incidence of 1.34/100 py. There were no significant differences between intervention versus control (1-way aHR: 1.19 [0.49 to 2.89], p = 0.70; 2-way aHR: 0.42 [0.13 to 1.37], p = 0.15).

Viral suppression results were similar in subgroup analyses stratified by age, phone sharing, pregnancy history, employment, HIV status disclosure, education, baseline VL, and baseline ART resistance (S1 Fig).

In exploratory analysis of MCH outcomes, we found significantly lower time to postpartum contraceptive initiation in 2-way versus control: 53.3% of participants in the control arm had initiated contraception by 6 months postpartum, compared with 58.0% of 1-way participants and 58.9% of 2-way participants (1-way aHR 1.09 [0.84 to 1.42], $p = 0.51$; 2-way aHR 1.31 [1.01 to 1.70], $p = 0.04$). No significant difference was found in time to cessation of EBF between arms; 80.2% of participants reported EBF to 6 months postpartum (79.3% of control participants, 81.6% of 1-way participants, and 79.6% of 2-way participants).

At exit, participants reported very high satisfaction with the intervention; 91.3% of 1-way participants and 94.2% of 2-way participants strongly agreed that they would recommend Mobile WACh-X to a friend. Participants reported finding the intervention helpful for HIV care (95.7% 1-way and 94.6% 2-way participants strongly agreed) and pregnancy care (97.0% 1-way and 95.4% 2-way participants strongly agreed) with >90% in both arms strongly agreeing that the intervention was helpful for providing new information, emotional support, and helping them take ART.

Overall, 121 adverse events were reported among 112 participants (13.5% of participants). These included maternal hospitalizations ($n = 9$), infant hospitalizations ($n = 14$), miscarriages ($n = 10$), stillbirths ($n = 33$), infant deaths ($n = 38$), maternal deaths ($n = 12$), and social harm ($n = 3$). The frequency did not differ significantly by arm. One event was determined to be potentially related to the intervention: an incident of intimate partner violence in which the participant's partner learned she was attending ANC.

## Discussion

In this 3-arm RCT, we found that during pregnancy to 2 years postpartum in PMTCT programs, 1-way and 2-way SMS did not significantly change maternal clinic engagement, retention, viral nonsuppression, or infant HIV-free survival. The previously published effect of this SMS platform on maternal postpartum contraceptive initiation was maintained. In exploratory analyses, 2-way SMS was associated with significantly increased viral suppression versus 1-way, consistent with evidence that bidirectional messages are more effective than unidirectional messages, though neither arm displayed higher viral suppression than control, and this finding should be interpreted with caution given the number of comparisons conducted [3]. Although WLWH and MCH providers endorse and enjoy SMS and telehealth services as noted in this and other studies [8,10], and numerous SMS platforms are being deployed for both HIV and MCH care [20], there are few RCTs rigorously assessing SMS effectiveness in improving clinical outcomes, and very few in the context of PMTCT [4]. To our knowledge, ours is the first study of a PMTCT SMS intervention to comprehensively assess maternal visit attendance, VL, resistance, and infant HIV-free survival with follow-up to 2 years postpartum. Our study demonstrates that mHealth systems engaging all mothers may not improve outcomes when on-time clinic attendance and viral suppression are already high in PMTCT programs.

Previous literature has demonstrated efficacy of some SMS interventions in improving HIV outcomes [1,2,21–23], Odeny and colleagues found that messages focused on visit attendance led to increased attendance and infant HIV testing in the first 8 weeks postpartum [24], though the effect was attenuated in a pragmatic trial [25]. Lester and colleagues found that open-ended "check-in" SMS with phone call triage of participant challenges improved viral suppression, but not retention, in adult HIV care [21,26]. Pop-Eleches and colleagues reported that weekly messaging is effective for ART adherence [27]. However, several other studies did not find significant impact on ART adherence [28,29] or viral suppression [30]. Differences in outcomes may be explained by differences in intervention content between studies, as well as

challenges detecting treatment effects in populations with high levels of adherence and viral suppression. We hypothesized that a platform which integrated HIV and MCH messaging would be effective in improving both HIV and MCH outcomes. The Mobile WACh MCH platform previously demonstrated significant benefit in prompt initiation of postpartum contraception and increased duration of EBF in HIV-uninfected women [12,13]. Based on input from formative interviews with mothers [8], and informed by the Health Belief Model and Social Cognitive Theory, Mobile WACh-X integrated ART adherence messages with messages on other priorities for WLWH, including facility delivery, EBF, infant immunization, and family planning. Mobile WACh-X also included options for covert or overt HIV-related messages per participant preference [10]. Our intervention had more HIV than MCH SMS (3:1), yet we found limited effect on HIV outcomes, while retaining effectiveness on contraception uptake. In the broader context of HIV care, it is unclear what messages or approaches reliably improve adherence and viral suppression, how to evaluate intervention mechanisms, or how to separate different intervention components [31–33]. Mobile WACh-X's lack of impact on engagement in HIV care may indicate that our messaging approach did not adequately target the salient constructs shaping this behavior in WLWH. We evaluated participants' ART-related information–motivation–behavior skills (IMB) using the LifeWindows IMB tool [34]. Women in this cohort had high IMB scores in all 3 domains; however, lower behavior skills scores were associated with viral nonsuppression at baseline [35]. Perhaps SMS grounded in the IMB skills theory and directly addressing behavior skills, or employing motivational interviewing approaches, would be more effective [36,37]. Additionally, it is likely that interest in repetitive adherence messaging declined over time, particularly when compared to MCH messages that addressed major changes in the peripartum/postpartum period with content that was fresh and relevant based on timing (pregnancy planning, delivery, and breastfeeding initiation of postpartum contraception).

In this study, overall retention in care, ART adherence, and viral suppression was high throughout the study, suggesting that most women, most of the time, did not have need for the intervention. It is therefore possible that the intervention would be more effective if it could facilitate timely identification and tailoring to the minority of women or the minority of times with low ART adherence behavioral skills, nonadherence, or unsuppressed VL [31]. There is increasing recognition that behavioral intervention efficacy is enhanced when interventions adapt dynamically to changes in participants' needs and contexts [38]. Mobile WACh-X messages were tailored for different "tracks" of women (primigravid or multigravid, adolescent or adult, and ART naïve or experienced) and times in the pregnancy and postpartum period, and 2-way nurse interactions were responsive to participants' questions. However, ART adherence message content was prespecified and delivered with minor adaptations over the pregnancy and 2-year postpartum period and did not incorporate links to VL data or ecological assessments of participant challenges. Our exploratory analysis comparing attendance of visits with and without a reminder SMS suggests better linkage between medical records and the SMS system may improve intervention efficacy, by delivering messages to the participants who need them at the times they need them. Among scheduled visits in the intervention arms, those that had a system-sent reminder SMS were attended more frequently than those who did not (90.2% versus 86.7%, $p < 0.001$), confirming that visit reminders were effective for those who received them. Linkage of SMS reminders with scheduled clinic visits required nurses to manually enter visit dates from medical records into the SMS system to trigger reminder messages. Challenges in locating paper records with the next appointment dates resulted in the system not always sending messages before the visit. This highlights challenges in administering SMS systems that are unlinked to medical records and the need to develop integrated systems to facilitate targeted interventions. Our study was implemented prior to wide use of electronic

medical records (EMRs) in MCH in Kenya. SMS systems integrated within EMRs are likely to improve visit attendance and provide additional data for precise message tailoring.

Few infants acquired HIV infection in this cohort, which is a testament to the success of PMTCT programs in Kenya. Overall, the MTCT rate was <1% during the 2-year follow-up, which is lower than anticipated but consistent with what is possible with PMTCT Option B + regimens [39,40]. Low MTCT occurred despite occasional maternal viral nonsuppression and resistance and may reflect infant prophylaxis in addition to generally good maternal ART adherence. Five percent of infants died during 2-year follow-up, underscoring continued high mortality in HIV-exposed infants and in the general population.

Strengths of our study include the study design, a multisite longitudinal RCT that included 2-year postpartum follow-up with detailed assessment of multiple important programmatic outcomes. We intentionally designed the study to evaluate programmatic outcomes to get a "real-world" perspective on the effectiveness of the intervention. While the landscape of mobile messaging is fast changing, and internet-based messaging and social media has grown exponentially in the last 5 years, we used SMS, a message format that is accessible on ubiquitous basic phones. Use of SMS means our study can be generalized to low-income and remote communities, and our findings are applicable across messaging platforms. Limitations included that abstraction of programmatic clinic and VL data posed challenges in completeness and data quality. During the study, there were changes in VL testing detection limits and in clinic medical records (transitioning in some cases from paper to electronic but with lapses back to paper at some sites due to power or internet outages). Visit reminder SMS required nurses to enter appointment dates from clinic records, which were not always available. Finally, high attendance, adherence, viral suppression, and infant HIV-free survival limited statistical power for some comparisons: our observed RRs from 0.80 to 1.36, with large confidence intervals (CIs), suggesting that effects may have been detected in a larger study.

In conclusion, this study found no significant impact of an integrated HIV/MCH SMS system on HIV outcomes, which may be explained by ineffective ART adherence messaging strategy or better than anticipated baseline PMTCT outcomes. There were few infant HIV infections despite maternal nonsuppression, and clinic attendance and ART adherence were high demonstrating excellent services in public sector Kenyan PMTCT programs during this period. Future SMS platforms for PMTCT may be more effective by targeting messaging to women at risk for nonadherence (based on behavioral skills or other indicators) or with demonstrated viral nonsuppression. Adaptive SMS platforms that integrate medical record and VL data can better tailor messaging that retains relevance to women in PMTCT. It remains important to rigorously evaluate SMS systems for effectiveness in improving clinical outcomes to ensure their effectiveness and optimally steward HIV care resources.

## Supporting information

**S1 CONSORT Checklist. CONSORT-recommended reporting of trial conduct.**
(DOC)

**S1 Table. Incidence rate of virologic nonsuppression by arm.**
(DOCX)

**S1 Fig. Subgroup analysis of effect of Mobile WAChX on viral nonsuppression.** ART, antiretroviral therapy; CI, confidence interval; GEE, generalized estimating equations; IMB, information–motivation–behavior skills; RR, relative risk; VL, viral load.
(TIF)

## Acknowledgments

We gratefully acknowledge participants in the Mobile WACh-X study as well as the Mobile WACh-X study team for their contributions to this work. We acknowledge our colleague, Brian Khasimwa, who was instrumental in executing this study and passed away before its completion. We acknowledge the University of Washington Global Center for Integrated Health of Women Adolescents and Children (Global WACh) for research support.

## Author Contributions

**Conceptualization:** John Kinuthia, Jennifer A. Unger, Alison L. Drake, Barbra A. Richardson, Grace John-Stewart.

**Data curation:** Keshet Ronen, Wenwen Jiang, Daniel Matemo, Trevor Perrier, Lusi Osborn.

**Formal analysis:** Keshet Ronen, Wenwen Jiang, Barbra A. Richardson, Grace John-Stewart.

**Funding acquisition:** John Kinuthia, Jennifer A. Unger, Alison L. Drake, Grace John-Stewart.

**Investigation:** Jennifer A. Unger, Wenwen Jiang, Daniel Matemo, Trevor Perrier, Lusi Osborn, Bhavna H. Chohan.

**Methodology:** Keshet Ronen, Jennifer A. Unger, Wenwen Jiang, Bhavna H. Chohan, Alison L. Drake.

**Project administration:** John Kinuthia, Keshet Ronen, Jennifer A. Unger, Daniel Matemo, Grace John-Stewart.

**Software:** Trevor Perrier.

**Supervision:** John Kinuthia, Grace John-Stewart.

**Validation:** Keshet Ronen, Barbra A. Richardson.

**Visualization:** Wenwen Jiang.

**Writing – original draft:** Keshet Ronen, Grace John-Stewart.

**Writing – review & editing:** John Kinuthia, Keshet Ronen, Jennifer A. Unger, Wenwen Jiang, Daniel Matemo, Trevor Perrier, Lusi Osborn, Bhavna H. Chohan, Alison L. Drake, Barbra A. Richardson, Grace John-Stewart.

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
