## [Editor Report · Decision Letter 0]

18 Feb 2021

Dear Dr John-Stewart, 

Thank you for submitting your manuscript entitled "SMS messaging to improve retention and viral suppression in prevention of mother-to-child HIV transmission (PMTCT) programs: a 3-arm randomized clinical trial" for consideration by PLOS Medicine.

Your manuscript has now been evaluated by the PLOS Medicine editorial staff and I am writing to let you know that we would like to send your submission out for external assessment.

Once your full submission is complete, your paper will undergo a series of checks in preparation for external assessment. 

Kind regards,

Richard Turner, PhD

rturner@plos.org

---

## [Decision Letter · Decision Letter 1]

7 Apr 2021

Dear Dr. Ronen,

Thank you very much for submitting your manuscript "SMS messaging to improve retention and viral suppression in prevention of mother-to-child HIV transmission (PMTCT) programs: a 3-arm randomized clinical trial" (PMEDICINE-D-21-00822R1) for consideration at PLOS Medicine. 

Your paper was discussed among the editors and sent to independent reviewers, including a statistical reviewer. The reviews are appended at the bottom of this email and any accompanying reviewer attachments can be seen via the link below:

[LINK]

In light of these reviews, we will not be able to accept the manuscript for publication in the journal in its current form, but we would like to invite you to submit a revised version that addresses the reviewers' and editors' comments fully. You will appreciate that we cannot make a decision about publication until we have seen the revised manuscript and your response, and we expect to seek re-review by one or more of the reviewers. 

We hope to receive your revised manuscript by May 3. Please email us (plosmedicine@plos.org) if you have any questions or concerns.

Please let me know if you have any questions, and we look forward to receiving your revised manuscript. 

Sincerely,

Richard Turner, PhD

rturner@plos.org

In the article metadata, it appears that the information under "competing interests" should be moved to the "funding" section. 

Please add "in Kenya" to the title, preceding the colon. 

Please quote study dates in your abstract.

Please combine the "Methods" and "Findings" subsections of your abstract; and add a new final sentence to the restructured combined subsection, which should begin "Study limitations include ..." or similar and quote 2-3 of the study's main limitations. 

Please quote the trial registration number with your abstract, but remove information on funding. 

After the abstract, please add a new and accessible "Author summary" section in non-identical prose. You may find it helpful to consult one or two recent research papers in PLOS Medicine to get a sense of the preferred style. 

Please remove trademarks, e.g., in your Methods section. 

Please remove "data not shown" from your Results, and incorporate the relevant data in supplementary files. 

Please remove information on competing interests and study funding (inc. grant numbers) from the end of the main text. In the event of publication, this information will appear in the article metadata, via entries in the submission form. 

Throughout the text, please style reference call-outs as follows: "... in Kenya [11,12]. We hypothesized ..." (noting the absence of spaces within the square brackets). 

In the reference list, please convert all boldface text and italics into plain text. Where appropriate, 6 author names should be listed rather than 3, followed by "et al.".

Does reference 18 lack a journal name? 

Please rename figure 1 "Participant flowchart" or similar. 

We understand that CONSORT discourages statistical comparisons of baseline characteristics, and ask you to remove this element from table 1. 

Please add a completed CONSORT checklist with your revision, labelled "S1_CONSORT_Checklist" or similar and referred to as such in your Methods section. In the checklist, please refer to individual items by section (e.g., "Methods") and paragraph number rather than by line or page numbers, as the latter generally change in the event of publication. 

Comments from the reviewers:

*** Reviewer #1: 

Statistical review

This paper reports a three arm randomised controlled trial testing SMS-based interventions for improving treatment of pregnant women with HIV. The trial was delivered successfully and was sizeable.

I had some comments on the statistical methods and reporting, listed below.

1. Abstract: I realise that if the abstract is required to be 300 words (I was not aware PLOS medicine had this as a hard limit), then it may be difficult to address some of the following issues as more words would be required.

- The authors mention an outcome being 'loss to follow up' - is this from the clinic or the trial?

- I would change 'no difference' to 'no significant difference' throughout. Since the trial was powered to testing the primary outcome it would be appropriate to add p-values in addition to the confidence intervals.

- "Median ART adherence (100%) and incident ART resistance (1.48/100 py) did not differ by arm." - does this mean that incident ART resistance was exactly 1.48/100 py in each arm? Otherwise I would drop the figure and put 'significantly differ' instead of differ.

2. Introduction, paragraph 2 - '1-way and 2-way arms' are used without much introduction (although it is briefly mentioned in the abstract).

3. Methods, randomisation and masking - this is well reported; can I just ask for the possible block sides to be added?

4. Methods, outcomes - I'd recommend the pre-specified subgroups are added here.

5. Methods, statistical analysis - when I first read this, I did not follow why the trial was powered on a composite criteria rather than the primary outcome itself. I think it is clearer once I read the protocol paper that this is separately the case for both outcomes. I would rewrite this a bit to ensure it is unambiguous.

6. Methods, statistical analysis - overall there seemed to be some differences in the planned analyses between the protocol paper and this paper. The biggest difference is that the primary outcome appeared to be described as a time-to-event outcome in the protocol paper (with the sample size calculation framing it in terms of a hazard ratio). I would recommend the authors describe differences from the protocol and reasons.

7. Methods, statistical analysis - was any adjustment for multiple testing, due to the three arms, made?

8. Methods, statistical analysis - what baseline characteristics were adjusted for?

9. Methods, statistical analysis - I didn't follow here (or perhaps it could be made clearer in the outcomes) whether the analysis was the proportion of eligible visits where the participant was non-suppressed, or the proportion of participants who were non-suppressed at any point. If the former I would recommend ensuring it is clear whether the arm assignment influenced the number of visits per participant.

10. Methods, statistical analysis - I didn't follow what the actual data for retention and engagement in care was. Is it a count variable of number of scheduled clinic visits kept? Were number of scheduled visits consistent across arms?

11. Results, table 1 - generally it is not recommended to test for significant differences in baseline characteristics. Here the number of significant variables is as expected given the number of characteristics considered.

12. Results, outcomes of RCT - it's not too clear what the quoted adjusted risk ratios refer to (see comment 9).

13. Results/discussion - I note the significant 2-way vs 1-way result is highlighted in the discussion. I would ensure it is clear that there were a lot of significance tests reported and this (marginally) significant result could be a false positive due to multiple testing.

14. Results, for the adherence data, when risk ratios are provided I'd clarify that this is referring to proportion of 'high adherence' periods.

15. Abstract/Methods/results - were there any serious adverse events/harms reported in the study? 

James Wason

*** Reviewer #2: 

This is a very well written and conducted evaluation of SMS for a particularly important HIV risk group, pregnant and post-partum women. The study is well reported and clearly well conducted and the findings are important. SMS is an increasingly popular method of communication with patient groups in Africa and is administered usually in a non-critical way. I think that this paper adds to the dialogue about recognizing both its utility and limitations. I have only a few important points, that I list below:

1) Perhaps in the introduction an again in the discussion there should be some recognition that in populations with high levels of the outcome of interest (say VL suppression), detecting treatment effects is difficult.

2) This was non-blinded rather than unblinded. 

3) Who actually administered the allocation to groups? What I mean is, when a patient has agreed to be in the study, how does the enrolling staff member get from consent to allocation?

4) I was a little surprised to see that the figures and tables were in the body of the manuscript. Did experienced writers see the final version?

5) In the discussion, I think you should add a section on limitations of this RCT and also limitations of our knowledge of how to evaluate communication tools with patients. The SMS literature is, in general, not that clear in whether it is a useful tool or not and yet it is widely used among health programs across many settings in Africa. I happen to believe it is useful (but that is just a belief not necessarily driven by the evidence). Could you discuss some the challenges of evaluating digital technologies in such a rapidly changing digital world? For example, you used SMS. But surely by the time this study was completed, people were using Whatsapp as their major form of communication? That's not a limitation, per se, but a challenge in such a rapidly moving environment. 

*** Reviewer #3: 

Overall, this was a good paper that was generally clearly written, and with a thoughtful discussion. However, the study size was not large, and greater critique of the assumptions made for the study power calculation would be helpful. 

Some specific comments, by section, are given below.

Abstract

1. Time on ART 1 year - does this mean follow-up from starting to 1 year after? Or on ART for 1 year at time of recruitment? It becomes clear in the main text, but needs to be clearer in the abstract.

2. Should report in the abstract the absolute values of the outcomes (percentages), and always by trial arm. For the purpose of the study, it is not so helpful to present figures pooled across trial arms and then say there was no difference by trial arm but not showing this. 

3. Related to the issue of study power, for which more detail should be provided in the methods of the main paper: the RR of 0.8 comparing 2-way SMS vs control for "ever virally non-suppressed" has a wide confidence interval, and it would be better to say "no evidence of a difference" rather than "no difference" compared to the control arm. I think this result does not correspond to the study power calculation, which was for the cumulative incidence of viral non-suppression by 24 months? It is important to present, as the first result, the one that corresponded to the primary study question for which the study was powered.

4. Need to quote figures on the time to postpartum contraception, especially as the conclusion says the SMS made a difference -but not quantified in the results. 

5. Not sure how the conclusion about targeted SMS follows from the results that are presented, the logic for this conclusion could be clarified in relation to the results presented.

Introduction

A bit more detail on the evidence to date on SMS to support adherence to ART post-partum would be helpful, beyond that there is "promising, but mixed evidence" and two papers cited. However, both the papers are systematic reviews - the introduction could mention that there are 2 systematic reviews, and then give a bit more information on what they found and whether there were patterns to the heterogeneity found e.g. did the SMS work better in particular settings, for particular groups of women, when implemented in particular ways?

Methods

1. Study power is very important here, given the study size is not large for a 3-arm trial and given the absence of evidence for effect on the primary outcomes. It seems from the methods text that the study was powered to be able to show a difference between 75% viral suppression in the control arm and 85% or more in the intervention arm, for a comparison of 2-year postpartum virologic non-suppression or retention. It could help to express this also as a risk ratio, given the analysis is done with risk (or rate) ratios, e.g. 15/25 = 0.6 so the study was powered to show a risk ratio of 0.6 (and then the risk ratios observed, were more of the order of 0.8 or 1/0.8 or closer to 1). 

Please clarify if this outcome was "virologic non-suppression at the 2 year time point" or "any virologic suppression during 2 years of follow-up". This matters, for which results correspond directly to the study power calculation. Please clarify exactly which analysis in the results corresponds to this power calculation (make sure the language in the results aligns precisely with the language for the study power calculation). 

2. Page 14 - says qualitative evaluation of participant experiences will be reported elsewhere, but something would be useful in this paper to explain lack of evidence of effect of the SMS. Please consider including some reference to what was learnt from the qualitative evaluation, in the discussion even if not in the results.

3. Page 15 - Please explain a bit more on "Anderson-Gill regression". Cumulative incidence could have been estimated using Kaplan-Meier methods and using Cox regression to compare trial arms; please explain how "Anderson-Gill regression" is different for a "time to event" analysis and why it was used instead of Kaplan-Meier estimates of the percentage with the outcome by different time points and Cox regression to estimate rate ratios. There is a footnote/caption to Figure 2 which explains why Anderson-Gill regression was used, and that it gave similar hazard ratios to Cox regression of "time to first event", but this information should be in the Methods.

Results

1. Please present first the results that correspond directly to the study power calculation. It is reported in the first paragraph of the "outcomes of RCT" paragraph that overall 20% were ever non-suppressed by 2 years - if this is the primary outcome, please report it in the text separately by trial arm. But, in Table 2 the focus is on (1) viral non-suppression at any time point (regardless of time point) and (2) viral suppression by particular time points i.e. 6, 12, 24 months post-partum. 

Please consider some re-ordering of the presentation of results, to first report on the primary outcomes defined in the study power calculation, and then to present other results that are informative.

2. Please clarify how the numerators and denominators are counted for outcome (2) [viral suppression by particular time points] in Table 2, as the model used is a log-binomial model; is the denominator for the 24-month time point all individuals who were followed up at 24 months? 

3. Confidence intervals are in general wide, e.g. aRR 0.80 [0.52-1.23], p=0.31 for comparing 2-way SMS with control for non-suppression over time (page 20). The wide confidence intervals, and how they relate to study power, could be given more attention in the discussion. 

4. Page 20 - Outcomes of RCT. Please report in the text the percentage with cumulative incidence of virologic suppression by 24 months, as well as the risk ratios. There are numerators and denominators for the log-binomial model used to estimate cumulative incidence of viral non-suppression in Table 2, but it is not completely clear how these numbers are counted - who contributes to the denominator, and who contributes to the numerator? - it is important that is clear in both Table 2 (e.g. with footnotes for more detail) and in the text. 

5. Was it considered to use the Kaplan-Meier method, rather than a log-binomial model, for a simple and transparent unadjusted analysis of the comparison of cumulative incidence of viral non-suppression by trial arm that allows for loss to follow-up? And then to use Cox regression to estimate hazard ratios to compare trial arms? It would help to understand the choice of the log-binomial model, if the numerators and denominators were precisely defined for each time point (who contributes to the analysis), see comment also above. 

Figure 2A shows "time to first non-suppression", I think from a Kaplan-Meier analysis, but then presents hazard ratios from Anderson-Gill regression that allow for repeated instances of viral non-suppression. I think it would be better for Figure 2A to show the hazard ratios from Cox regression, that directly correspond to the graph. The findings from the Anderson-Gill regression could be included in a table instead - they should be included as part of Table 2, and I think they correspond closely to the primary analysis defined for the study power calculation (cumulative incidence of viral suppression by 24 months) but are not exactly the same as they allow for repeated events (of viral non-suppression).

6. In Table 2, please write out what RR stands for in the table column heading, as it matters for interpretation whether is a risk ratio (which I think it is) or a rate ratio and it needs to be clear for the reader and correspond to the analysis that was done. I think from the log-binomial models, it is a risk ratio, but for example if Cox regression were done it would be a rate ratio / hazard ratio.

***

[LINK]

---

## [Decision Letter · Decision Letter 2]

6 May 2021

Dear Dr. Ronen,

Thank you very much for re-submitting your manuscript "SMS messaging to improve retention and viral suppression in prevention of mother-to-child HIV transmission (PMTCT) programs in Kenya: a 3-arm randomized clinical trial" (PMEDICINE-D-21-00822R2) for consideration at PLOS Medicine.

I have discussed the paper with editorial colleagues and it was also seen again by two reviewers. I am pleased to tell you that, provided the remaining editorial and production issues are dealt with, we expect to be able to accept the paper for publication in the journal.

[LINK]

Please let me know if you have any questions, and we look forward to receiving the revised manuscript shortly.   

Sincerely,

Richard Turner, PhD

rturner@plos.org

Requests from Editors:

At line 19 (abstract), please make that "infant HIV infection". 

At line 22, please adapt the text to "... but was associated with improved initiation of postpartum contraception" or similar, as this result is apparently from an exploratory analysis.

At line 23, we suggest "women are virally suppressed". 

At line 27 (author summary), please make that "previously reported apparent effect" or similar, as noted above. 

Please make randomized/randomised and derivatives consistent through the paper. 

Please remove the competing interest statement from the end of the main text.

Please make that "PLoS ONE" in reference 29.

Please adapt the attached CONSORT checklist so that individual items are referred to by section (e.g., "Methods") and paragraph number. Page and line numbers should not be used, as these generally change in the published paper. 

Comments from Reviewers:

*** Reviewer #1: 

Thank you to the authors for addressing all my previous comments well. I have no further issues to raise.

*** Reviewer #2: 

The authors have done a very good job addressing all of the reviewer's comments and suggestions. They should be commended for the hard work necessary to conduct and successfully complete this trial.

***

[LINK]

---

## [Editor Report · Decision Letter 3]

9 May 2021

Dear Dr Ronen, 

On behalf of my colleagues and the Academic Editor, Dr Mills, I am pleased to inform you that we have agreed to publish your manuscript "SMS messaging to improve retention and viral suppression in prevention of mother-to-child HIV transmission (PMTCT) programs in Kenya: a 3-arm randomized clinical trial" (PMEDICINE-D-21-00822R3) in PLOS Medicine.

Prior to final acceptance, please:

Make that "... through health record integration" in the author summary, and

Noting our submission guidelines (https://journals.plos.org/plosmedicine/s/submission-guidelines), please remove "(Chohan, submitted)" from your Discussion section, and if necessary remove the relevant sentence. 

PRESS

Sincerely, 

Richard Turner, PhD 

rturner@plos.org